# Classification of Skin Cancer Lesions Using Explainable Deep Learning

**DOI:** 10.3390/s22186915

**Published:** 2022-09-13

**Authors:** Muhammad Zia Ur Rehman, Fawad Ahmed, Suliman A. Alsuhibany, Sajjad Shaukat Jamal, Muhammad Zulfiqar Ali, Jawad Ahmad

**Affiliations:** 1Department of Electrical Engineering, HITEC University Taxila, Taxila 47080, Pakistan; 2Department of Cyber Security, Pakistan Navy Engineering College, National University of Sciences & Technology, Karachi 75350, Pakistan; 3Department of Computer Science, College of Computer, Qassim University, Buraydah 51452, Saudi Arabia; 4Department of Mathematics, College of Science, King Khalid University, Abha 61413, Saudi Arabia; 5James Watt School of Engineering, University of Glasgow, Glasgow G12 8QQ, UK; 6School of Computing, Edinburgh Napier University, Edinburgh EH10 5DT, UK

**Keywords:** classification, deep learning, explainable AI (XAI), skin cancer, transfer learning

## Abstract

Skin cancer is among the most prevalent and life-threatening forms of cancer that occur worldwide. Traditional methods of skin cancer detection need an in-depth physical examination by a medical professional, which is time-consuming in some cases. Recently, computer-aided medical diagnostic systems have gained popularity due to their effectiveness and efficiency. These systems can assist dermatologists in the early detection of skin cancer, which can be lifesaving. In this paper, the pre-trained MobileNetV2 and DenseNet201 deep learning models are modified by adding additional convolution layers to effectively detect skin cancer. Specifically, for both models, the modification includes stacking three convolutional layers at the end of both the models. A thorough comparison proves that the modified models show their superiority over the original pre-trained MobileNetV2 and DenseNet201 models. The proposed method can detect both benign and malignant classes. The results indicate that the proposed Modified DenseNet201 model achieves 95.50% accuracy and state-of-the-art performance when compared with other techniques present in the literature. In addition, the sensitivity and specificity of the Modified DenseNet201 model are 93.96% and 97.03%, respectively.

## 1. Introduction

According to World Health Organization (WHO) statistics, skin cancer accounts for one-third of all reported cancer cases, and the prevalence rate is increasing globally [1]. Over the past decade, a prominent increase in skin cancer has been reported in the USA, Australia, and Canada. Approximately 15,000 people do not survive every year after being infected by skin cancer [2]. An American study shows that 7180 people died in 2021 of only one type of cancer and it is expected that in the year 2022, nearly 7650 people will die because of melanoma cancer [3]. Depletion of the ozone layer increases the amount of hazardous ultraviolet (UV) radiation reaching the surface of earth. The UV radiations can damage skin cells, which may lead to the cancerous growth of cells. The hazardous UV radiation can cause a wide range of adverse consequences, and one of them is the growing incidence rate of skin cancer [4]. Some other factors including smoking, usage of alcohol, different infections, viruses, and the living environment also trigger the growth of cancerous cells. There are two major types of skin tumors; some tumors are cancerous while others are non-cancerous. The malignant tumor is cancerous, and it has further types [5]. The most prevalent malignant skin lesions are squamous cell cancer (SCC), basal cell cancer (BCC), malignant melanoma, and dermatofibrosarcoma. Numerous more include malignant cutaneous adnexae, fibrous histiocytomas, Kaposi’s sarcoma, and pleomorphic sarcoma [5,6,7]. Malignant melanoma is rare skin cancer but is considered the deadliest cancer of all of them. Malignant tumors spread to other body organs through the lymphatic system or blood vessels; this spread is called metastasis [8].

Abnormal growth of melanocytic skin cells causes malignant melanoma. Exposure of skin to sunlight produces melanin, which naturally protects the skin from adverse effects of sunlight, however, if melanin is accumulated, the tumor starts to develop [9]. In many situations, total excision (surgical removal) of an infected region performed by surgeons results in health. Moreover, in several cases, it requires rehabilitation of the infected region that is performed by plastic surgeons [7]. On the other hand, the benign tumor is non-cancerous and it does not spread to other organs but has the capability to enlarge lesion patches and tumors. Different types of benign tumors include seborrheic keratoses, cherry angiomas, dermatofibromas, skin tags (acrochordon), pyrogenic granulomas, and cysts [5,10].

In order to diagnose skin cancer, a series of steps are performed by a physician. In the first stage, lesions are inspected by the naked eye. For further examination, dermoscopy is used to further examine the pattern of skin lesions. In this process, a gel is applied to the visible skin lesions and examined under a magnifying tool for better visualization [11]. For more detailed analysis, a part of the suspected region of skin is removed and sent to a lab for microscopic examination; this procedure is referred to as a biopsy. Some experts diagnose the skin lesion based on the ABCDE technique in which a number of factors are analyzed, including color, border, asymmetry, the diameter of lesion, and development of lesion over time [12]. However, the inspection solely depends on the skill of the dermatologists along with the clinical facilities. Timely detection and diagnosis of cancer, specifically skin cancer, can avoid further spread and be cured effectively. Moreover, early detection of skin cancer helps to reduce the mortality rate and expensive medical procedures [13]. Furthermore, the manual procedure for the inspection of skin cancer is time-consuming and there could be a chance of human error during the process of diagnosis.

Since the last decade, the use of computer-aided systems has been seen in the field of medicine. Such systems can be used for the detection of skin cancer. Traditionally, different skin-related features such as color, texture, shape, etc., have been used [14]. Extraction of multiple features is not only time-consuming but also a complex process. However, recent developments in the field of artificial intelligence have paved the path for feature extraction using deep learning architectures. Deep learning architectures can extract multiple features using convolutional neural networks (CNNs) [15]. CNN can extract features efficiently as compared to traditional methods of feature extraction. Recently, deep learning-based computer-aided systems have been used for the diagnosis of different diseases and have shown remarkable results. There is a huge potential for using computer-aided systems to aid medical staff in disease diagnosis in its early stages.

## 2. Related Work

Recently numerous schemes have been developed for the classification of skin lesions using deep learning architectures. Some of the recent techniques for the classification of skin lesions are discussed in this section.

Dorj et al. [16] used a pre-trained AlexNet model for the process of feature extraction, while SVM was used for classification. The technique for the classification of skin lesions yielded impressive results. Filho et al. [17] presented a technique for skin lesion classification using a structural co-occurrence matrix (SCM). The SCM is used to extract texture features from dermoscopic images. Experimentation was performed on the ISIC 2016 and ISIC 2017 datasets. For classification, various learning algorithms were used, and among them, the SVM gave the best results. This technique attained a specificity of 90%. Li et al. [18] proposed a novel lesion indexing network for the classification of skin lesions. The LIN is made up of a deep learning algorithm that is capable of extracting additional features as compared to a simple deep learning algorithm. The proposed scheme attained good classification results with an accuracy of 91.2%. The scheme can also segment lesions, however, the results needed significant improvement for the segmentation of lesions. Saba et al. [19] have proposed a deep learning-based approach for the recognition of skin cancer. To improve the visual quality of the used datasets, a contrast stretching technique was used. Moreover, to estimate the boundary of lesions, a CNN followed by XOR operation was used. The features are extracted using Inceptionv3 with the help of transfer learning. For testing, the PH2 and ISIC 2017 datasets were used.

Esteva et al. [20] proposed a technique for the classification of skin cancer using the inceptionV3 model. Clinical images were used for training and evaluation purposes. The results of the proposed technique were cross-examined by a 21-member certified board for the two most deadly skin cancers. Le et al. [21] presented a deep learning technique based on the ResNet50 model that uses a transfer learning approach for model training. The hyperparameters were fine-tuned to improve the performance of the model. Moreover, to avoid overfitting, instead of the average pooling layer, global average pooling layers were used. The HAM10000 dataset was used in this work. Iqbal et al. [22] presented a technique for skin lesion classification based on deep convolutional neural networks. The model used in this work consists of multiple blocks in a top to bottom formation, which provides feature information at different scales. The model comprises 68 convolutional layers, and the ISIC 2017–2019 dataset was used in this work. Srinivasa et al. [23] proposed a technique that utilizes MobileNetV2 and LSTM networks. The proposed technique is used for lesion classification, which is performed on the HAM1000 dataset. MobileNetV2 is a lightweight model that requires low computational power and is adaptable to end devices. The LSTM network is employed to preserve temporal details of features extracted from the MobileNetV2. The combination of LSTM with MobileNetV2 improved the accuracy up to 85.34%.

Shahin et al. [24] proposed a deep learning technique based on deep convolutional neural networks for automated skin cancer classification. The proposed technique can perform binary classification for skin cancers for both benign and malignant cases. Different preprocessing steps include noise removal, normalization, and data augmentation. Several deep learning models were compared in this work to attain better and more effective classification accuracy. The proposed technique attained a test accuracy of 91.93% on the renowned HAM10000 dataset. Farhat et al. [25] presented a methodology for the classification of skin cancer employing a deep learning algorithm. Two different datasets have been used, which are HAM10000 and ISIC 2018. The proposed technique comprises a number of steps, of which the major steps are feature extraction using the deep learning model and feature selection using a metaheuristic algorithm. The final classification was performed using extreme machine learning with an accuracy of 93.40% and 94.36% for HAM10000 and ISIC2018, respectively. Chaturvedi et al. [26] proposed a multi-class skin classification technique based on deep learning models. The automated deep learning-based system tends to improve classification accuracy. Deep learning models are fine-tuned to improve the accuracy of models; moreover, ensemble models have also been used for comparison. The technique attained an accuracy of 93.20% on the HAM10000 dataset.

## 3. Materials and Methods

A new framework for the classification of skin lesions is presented in this section. The proposed framework can differentiate cancerous and non-cancerous lesions using deep learning models. The proposed framework requires a series of steps for the efficient classification of lesions. It starts by augmenting the available dataset and subsequently follows steps to retrain the deep learning model that includes transfer learning, fine-tuning of the model along with hyperparameter tuning. The fine-tuned model is able to extract desired features for the classification of skin cancer lesions. Two different deep learning techniques are used in this work. The augmented dataset is used for the fine-tuned deep learning model according to the requirements of this work. The general workflow of the proposed framework is presented in Figure 1. Each step of the workflow is discussed in the following subsections.

### 3.1. Dataset and Data Preprocessing

The dataset is the primary requirement for using deep learning models for different problems. The datasets are not readily available, and moreover, clean and well-prepared datasets are rare. Deep learning models learn the patterns and features of the dataset, and based on the learned patterns and features, they can perform predictions. A clean and well-prepared dataset is the key requirement to attaining state-of-the-art performance using deep learning models. In this work, the dataset is acquired from Kaggle [27]. Kaggle is a well-known platform for the scientific community; the dataset is part of ISIC archive [28]. The dataset consists of two categories: malignant and benign. A total of 3297 images are present in this dataset. The category “benign” contains 1800 images, while “malignant” contains 1497 images.

Data preprocessing is a vital step in enhancing the quality of any dataset [29]. It consists of multiple approaches; in this work, the visual quality of the dataset is enhanced using contrast stretching. It significantly improves the quality of images where lesion spots are faded during the process of image acquisition. A few contrasted enhanced images are shown in Figure 2.

Secondly, deep learning models require a large amount of data for training [30]. To increase the training examples of the dataset, data augmentation is used. Different data augmentation techniques are present in the literature for different scenarios. Three different augmentation techniques have been used in this work, rotation, flipping, and noise addition. Rotation and flipping are scale-invariant augmentation operations. Images are rotated at 15 and 45 degrees in both clockwise and anti-clockwise directions followed by horizontal flipping [31]. The last augmentation technique used in this work is noise addition, which is commonly known as noise augmentation. This tends to inject random noise into the dataset to increase the number of samples present in the dataset. This technique not only enlarges the dataset but also reduces the generalization error during the process of training deep learning model, making the process of training robust [32]. The Gaussian noise has been added to the dataset with a variance of 0.1. A detailed description of the dataset is presented in Table 1. A few samples of the augmented dataset are shown in Figure 3.

### 3.2. Deep Learning Models

Deep learning is the subfield of artificial intelligence (AI), which mimics the behavior of a human brain. During the last decade, it comes up in the spotlight when it has been used in different fields for different purposes and provided superior results compared to other existing algorithms. It is called “deep learning” as it is composed of a large number of hidden layers that are usually convolutional layers [33]. These hidden convolution layers are being used for feature extraction. Deep learning models are a step toward the automation of computer-aided systems. These models have been used for various purposes including classification, segmentation, object detection, etc., in different fields, such as agriculture, medical, driverless cars, and many more.

In this work, two deep learning models have been used for the classification of skin diseases. The used models are MobileNetV2 [34] and DenseNet201 [35]. These models are trained using Transfer Learning (TL). The TL approach not only aids the model in feature learning but also improves performance while limiting computation resources [36]. Both models have undergone minor modifications in order to achieve the desired outcomes. The used models are discussed below, along with the modifications.

#### 3.2.1. Modified MobileNetV2

One of the models used in this work is MobileNetV2, which is a well-known model for feature extraction. Being a lightweight model, it is extensively used in the research domain. The pretrained MobileNetV2 model used in this work has been previously trained on a large image dataset, the ImageNet [37]. In this work, transfer learning is used to train the pretrained MobileNetV2 model. The original MobileNetV2 model takes the image of size 224 × 224 × 3. The image is passed through a convolutional layer having 32 filters. The inverted residual block (IRB) is the predominant block of MobileNetV2, which reduces the memory requirement as compared to the usual convolutional blocks. The IRB consists of point-wise and depth-wise convolution. The depth-wise convolution is used to eliminate redundant feature; the elimination of redundant features helps the model perform better while maintaining a low computational cost. The ReLu6 has been used as an activation function throughout the network [34]. The IRB of MobileNetV2 is commonly referred to as bottleneck; there are 17 such bottlenecks present in MobileNetV2. Figure 4 depicts the architecture of the proposed modified MobileNetV2.

In this work, three 2D convolution layers are stacked at the end of the network, which improves the performance of the model significantly. The layer CONV1_1 contains 128 filters with a kernel size of 3 × 1, the layer CONV_2 also contains 128 filters having a kernel size of 3 × 1. The final convolution layer CONV_2 contains 64 filters with a kernel size of 3 × 3. The classification head is used for final classification and is composed of a GAP layer, a batch normalization layer, and two dense layers. The final dense layer is modified according to the desired requirements.

#### 3.2.2. Modified DenseNet201

The other model used for feature extraction is the pre-trained DenseNet201, where the number 201 refers to 201 layers of the original model. The DenseNet201 model is also trained on the ImageNet dataset. Transfer learning is used to train the model for the desired task of classifying skin diseases. The DenseNet201 model consists of 4 dense blocks and 3 transition layers that act as connections between the two dense blocks. Inside the dense block, each convolution layer is connected to other convolution layers. After every dense block, the size of the feature map is increased. The transition layers act as the downsampling layers. In DenseNet201, downsampling is performed using average pooling [35]. The architecture of the proposed Modified DenseNet201 is shown in Figure 5.

Some convolution layers are also stacked up at the end of the fourth dense block. The convolutional layer, namely the CONV1_1 layer is composed of 128 filters with a kernel size of 3 × 1 and the convolution layer CONV1_2 contains 128 filters having a kernel size of 1 × 3. The purpose of decomposing 3 × 3 kernel into 3 × 1 and 1 × 3 kernel is to reduce computational power. The third convolution layer, CONV_2 is built up with 64 filters with a kernel size of 3 × 3. The extracted features are fed into the classification head which comprises of GAP, a batch normalization layer, 2 dense layers, and the Softmax classifier. Table 2 outlines the architecture of the proposed Modified DenseNet201.

### 3.3. Grad-CAM Visualization

In this work, Gradient-weighted class activation mapping (Grad-CAM) visualization is used to get an insight into feature learning using the deep learning model. Generally speaking, deep learning models are considered as black boxes, as they take input and give their predictive output. Grad-CAM visualization has been used recently to understand what is happening inside the deep learning model. An illustration of Grad-CAM visualization is shown in Figure 6. It is a weakly supervised localization technique.

## 4. Results

The results of the proposed technique are presented in this section. Section 4.1 provides brief information on the experimental setup. The results using MobileNetV2 and DenseNet201 are presented in Section 4.2 and Section 4.3, respectively. Results are analyzed and compared in Section 4.4.

### 4.1. Experimental Setup

The dataset used in this work is taken from Kaggle and consists of two classes. The dataset is divided such that 70% is used for training, 20% for validation and the remaining 10% for testing. Google Colab has been used for running the MobileNetV2, DenseNet201 and their proposed modified models.

### 4.2. Results Based on MobileNetV2

The results using the MobileNetV2 and proposed modified MobileNetV2 are shown in Table 3. The MobileNetV2 attains an accuracy of 90.54% when trained on the dataset. The sensitivity and specificity achieved were 89.93% and 91.18%, while precision and sensitivity and F1 scores were 91.32% and 90.62%, respectively. Results of the proposed Modified MobileNetV2 are also presented in Table 3, which shows that the Modified MobileNetV2 attains an accuracy of 91.86%. Other parameters, including sensitivity, specificity, precision, and F1 scores are recorded as 91.09%, 92.66%, 92.82%, and 91.95%, respectively. Moreover, the results presented in Table 3 are also validated through the confusing matrix that is presented in Figure 7.

Figure 7a presents the confusion matrix of MobileNetV2 while Figure 7b represents the results of the proposed Modified MobileNetV2 using the confusion matrix. Figure 7a shows that MobileNetV2 model is capable of detecting benign class with an accuracy of 89%, while 11% of benign class was classified as malignant. The malignant class was accurately detected with 91% accuracy. Similarly, Figure 7b shows that the proposed Modified MobileNetV2 model detects the benign class with an accuracy of 91% and 9% of benign class was misclassified as malignant. In addition, the malignant class attains a detection accuracy of 93%, while only 7% of the malignant class was detected as benign. It can be seen from Table 3 and Figure 7 that the proposed Modified MobileNetV2 showed better performance based on the evaluation parameters. The accuracy and loss plots of the proposed Modified MobileNetV2 are presented in Figure 8.

### 4.3. Results Based on DenseNet201

This subsection presents the results attained using the DenseNet201 and the proposed Modified DenseNet201. The detailed results are presented in Table 4, which is based on several evaluation parameters. Table 4 shows that DenseNet201 attained an accuracy of 94.09% on the used dataset. In addition, the sensitivity, specificity, precision, and F1 score were recorded as 92.16%, 96.05%, 95.96%, and 94.02%, respectively. The proposed Modified DenseNet201 shows superiority over the pre-trained DenseNet201 by attaining an accuracy of 95.50%. To further ensure the authenticity of the obtained results, several other parameters were also considered. The sensitivity and specificity obtained by the Modified DenseNet201 model are 96.96% and 97.06%, respectively. Whereas precision and F1 score were recorded as 97.02% and 95.46%, respectively. The results shown in Table 4 are also verified using the confusion matrix shown in Figure 9.

The confusion matrix of DenseNet201 is shown in Figure 9a, whereas Figure 9b shows the confusion matrix of the proposed Modified DenseNet201. Figure 9a shows that Benign disease was correctly classified with an accuracy of 92%, while only 8% of the Benign disease was misclassified as Malignant, whereas according to the confusion matrix, the malignant disease was classified with an accuracy of 96%. In this case, the malignant disease was only 4% misclassified as Benign. Figure 9b presents the confusion matrix of the proposed Modified DenseNet201. The confusion shows that Benign and Malignant diseases were accurately classified with an accuracy of 94% and 97%, respectively. Only 6% of the benign disease was misclassified as malignant. The accuracy and loss plots of the proposed Modified DenseNet201 are shown in Figure 10.

Furthermore, a visual illustration of lesion spots is detected using the Grad-CAM technique. As discussed, the Grad-CAM weakly localized the lesion spot that can certainly aid the medical staff in detection and diagnosis. Using the proposed technique, it is evident that the learned features are well trained and are able to detect and localize lesion spots based on feature information. Figure 11 shows a few sample images that visually illustrate the purpose of Grad-CAM. The samples shown below are the results of the proposed Modified DenseNet201 with an accuracy of 95.50%. Figure 11a shows the original image, while Figure 11b shows its respective Grad-CAM based localization of the lesion spot.

### 4.4. Analysis and Comparison

The results in this subsection are analyzed by comparing the results of the proposed technique with the original pre-trained models. Moreover, the results are also compared with other techniques present in the literature used for the classification of skin cancer.

The accuracies attained using different models used in this work are shown in Figure 12. It is observed that the proposed Modified MobileNetV2 and Modified DenseNet201 performed better in comparison to the original pre-trained MobileNetV2 and DenseNet201 models. Moreover, as illustrated in Figure 12, the proposed Modified DenseNet201 outperforms the other three models. The proposed technique is compared with other techniques present in the literature for the classification of skin cancer. Table 5 shows that the proposed technique demonstrates its superiority over other techniques by successfully attaining 95.5% accuracy.

## 5. Conclusions

In this paper, the pre-trained MobileNetV2 and DenseNet201 deep learning models were modified by adding additional convolution layers to effectively detect skin cancer. Specifically, for both models, the modification includes stacking three convolutional layers at the end of both models. In addition, the classification head was modified by employing a batch normalization layer, and the final classification layer was also modified according to the class of problem under consideration. Experiments indicate that the performance of both models was increased following the architectural modifications. The Modified DenseNet201 gave the highest accuracy as compared to the other three models used in this study. The proposed Modified DenseNet201 model can be used for multi-class skin cancer diagnosis with slight changes. In addition, the optimization strategies available in the literature can be utilized for improved results.

## Figures and Tables

**Figure 1 sensors-22-06915-f001:**
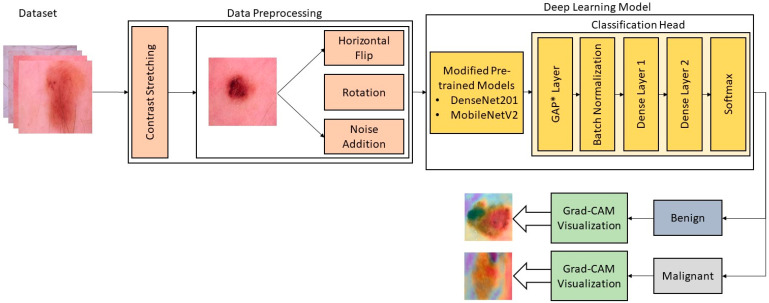
The workflow of the proposed technique for skin cancer classification.

**Figure 2 sensors-22-06915-f002:**
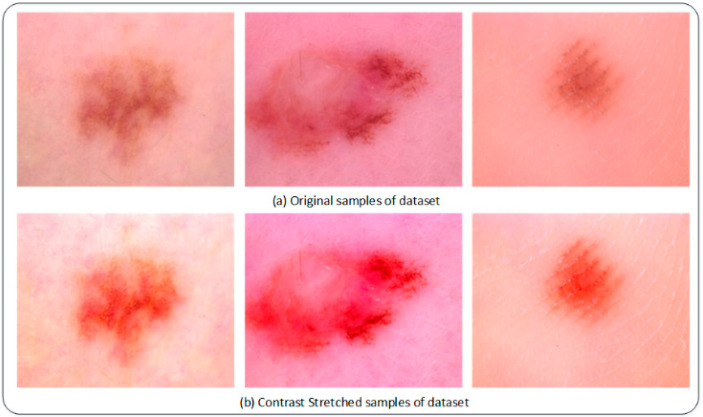
Illustration of contrast-enhanced image.

**Figure 3 sensors-22-06915-f003:**
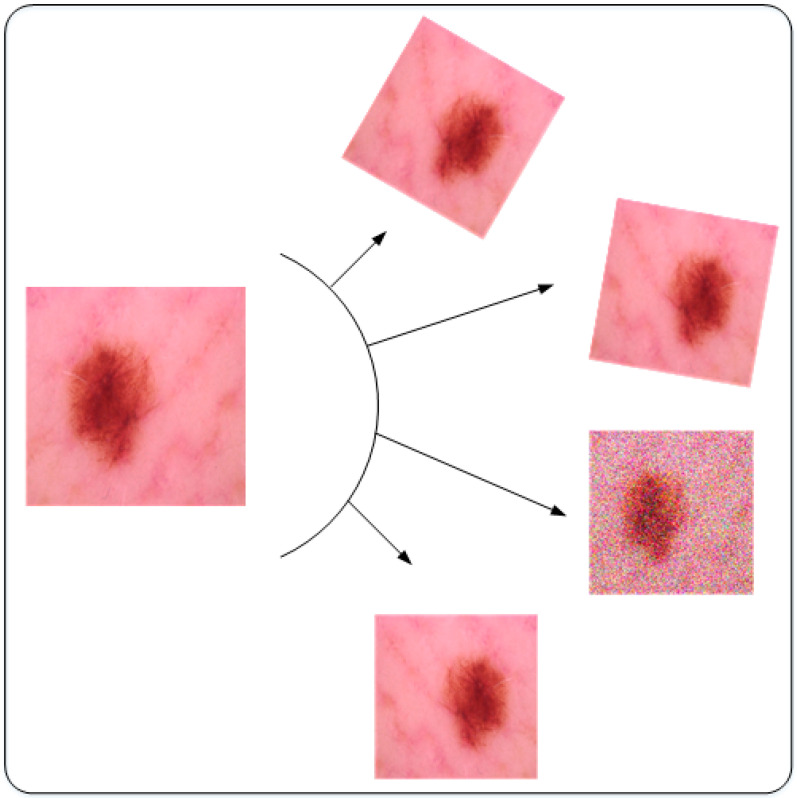
Visual illustration of the augmentation techniques.

**Figure 4 sensors-22-06915-f004:**
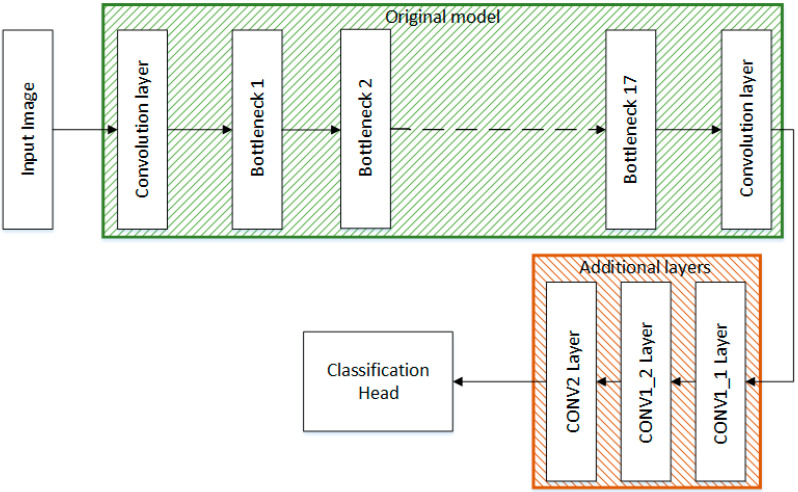
Architecture of the proposed modified MobileNetV2.

**Figure 5 sensors-22-06915-f005:**
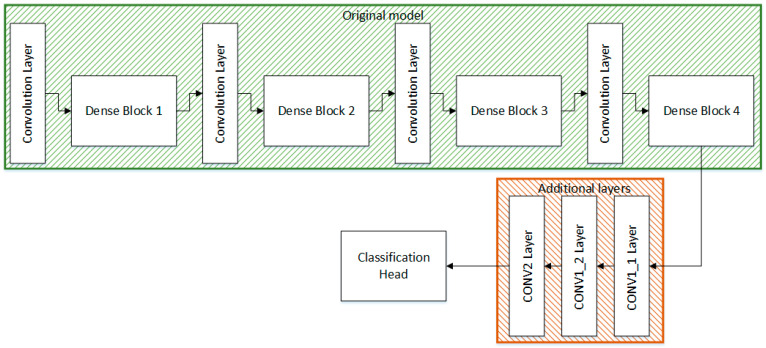
Architecture of the proposed Modified DenseNet201.

**Figure 6 sensors-22-06915-f006:**
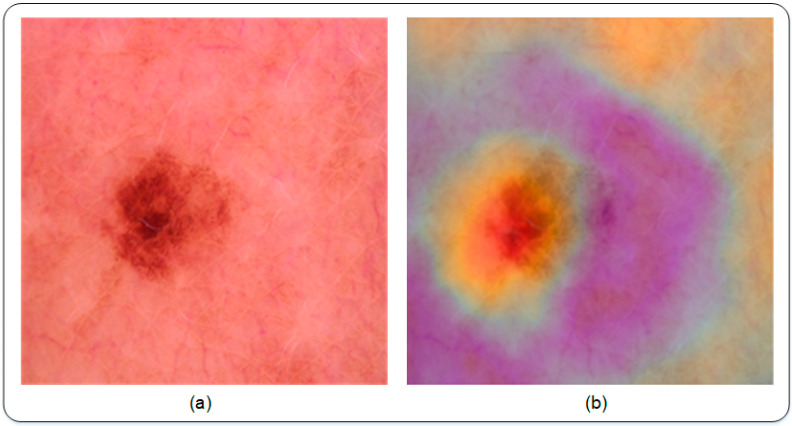
A visual illustration of localization using Grad-CAM. (**a**) shows the input image and (**b**) depicts the lesion being weakly localized in the image using Grad-CAM. The result presented in (**b**) demonstrates the fact that Grad-CAM is capable of localizing lesion present in the image.

**Figure 7 sensors-22-06915-f007:**
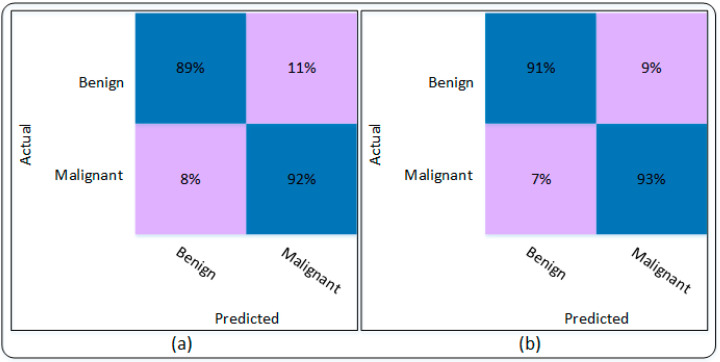
Confusion matrix of the proposed technique; (**a**) MobileNetV2 model, (**b**) Modified MobileNetV2 model.

**Figure 8 sensors-22-06915-f008:**
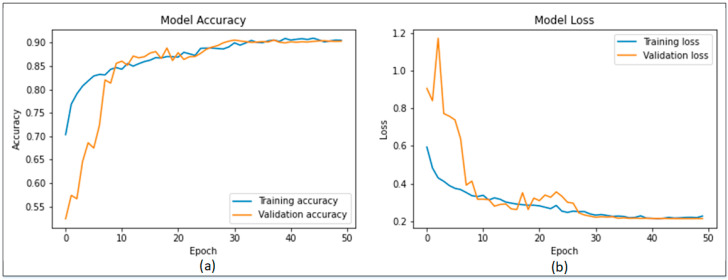
Training plots of Modified MobileNetV2; (**a**) accuracy plot, (**b**) loss plot.

**Figure 9 sensors-22-06915-f009:**
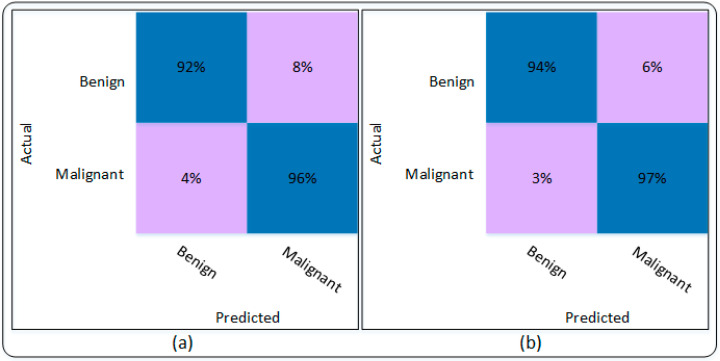
Confusion matrix of the proposed technique. (**a**) DenseNet201 model, (**b**) Modified DenseNet201 model.

**Figure 10 sensors-22-06915-f010:**
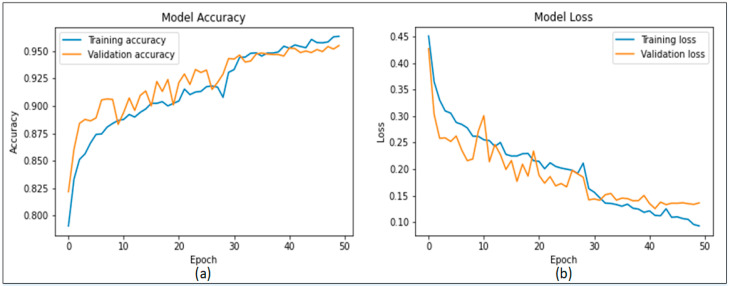
Training plots of Modified DenseNet201 (**a**) Accuracy plot (**b**) Loss plot.

**Figure 11 sensors-22-06915-f011:**
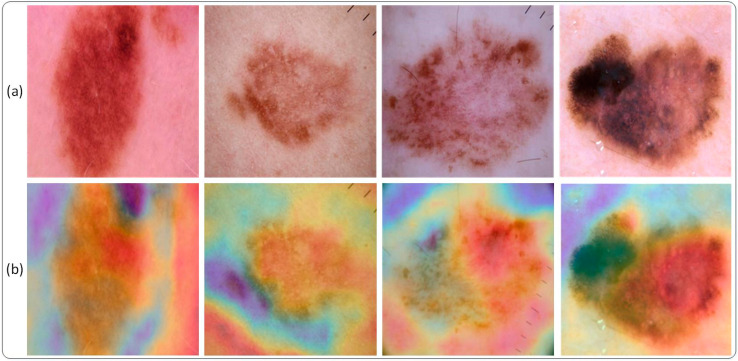
An Illustration of detection of lesion spots using Modified DenseNet201 based on Grad-CAM (**a**) shows the input image and (**b**) depicts the final output of Modified DenseNet201 using Grad-CAM.

**Figure 12 sensors-22-06915-f012:**
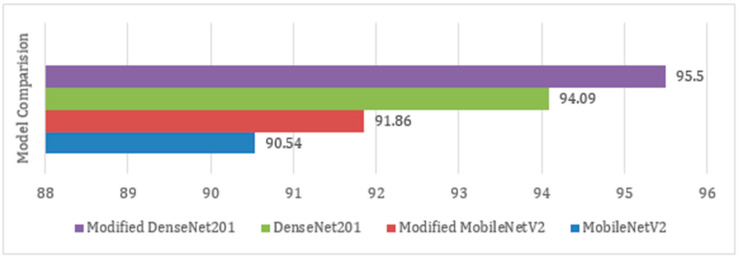
Performance comparison of the four models used in this work.

**Table 1 sensors-22-06915-t001:** A detailed description of the dataset.

Categories	Original Images	Augmented Images	Training Images	Validation Images	Testing Images
Benign	1800	3727	2609	745	373
Malignant	1497	3600	2520	720	360
Total	3297	7327	5129	1465	733

**Table 2 sensors-22-06915-t002:** Architecture of the proposed Modified DenseNet201.

Layers	DenseNet201
Convolution	7 × 7 *conv*, stride 2
Pooling	2 × 2 max pool, stride 2
Dense block (1)	$\left[\begin{array}{c} 1\times 1\;conv\\ 3\times 3\;conv \end{array}\right]\times 6$
Transition layer (1)	1 × 1 *conv*
	3 × 3 max pool, stride 2
Dense block (2)	$\left[\begin{array}{c} 1\times 1\;conv\\ 3\times 3\;conv \end{array}\right]\times 12$
Transition layer (2)	1 × 1 *conv*
	2 × 2 average pool, stride 2
Dense block (3)	$\left[\begin{array}{c} 1\times 1\;conv\\ 3\times 3\;conv \end{array}\right]\times 48$
Transition layer (3)	1 × 1 *conv*
	2 × 2 average pool, stride 2
Dense block (4)	$\left[\begin{array}{c} 1\times 1\;conv\\ 3\times 3\;conv \end{array}\right]\times 32$
CONV1_1	3 × 1 *conv*, filters 128
CONV1_2	1 × 3 *conv*, filters 128
CONV2	3 × 3 *conv*, filters 64
Classification Layer	Global Average Pooling
	Classification Head

**Table 3 sensors-22-06915-t003:** Classification results using MobileNetV2.

Deep Learning Model	Accuracy	Sensitivity	Specificity	Precision	F1 Score
MobileNetV2	90.54%	89.93%	91.18%	91.32%	90.62%
Modified MobileNetV2	91.86%	91.09%	92.66%	92.82%	91.95%

**Table 4 sensors-22-06915-t004:** Classification results using Modified DenseNet201.

Deep Learning Model	Accuracy	Sensitivity	Specificity	Precision	F1 Score
DenseNet201	94.09%	92.16%	96.05%	95.96%	94.02%
Modified DenseNet201	95.50%	93.96%	97.06%	97.02%	95.46%

**Table 5 sensors-22-06915-t005:** Comparison with state of the art techniques.

References	Accuracy	Year
Srinivasa et al. [23]	85.34%	2021
Shahin et al. [24]	91.93%	2021
Farhat et al. [25]	94.36%	2022
Proposed	95.50%	-

## Data Availability

The dataset used in this research is publicly available with the name “Skin Cancer: Malignant vs. Benign”, on https://www.kaggle.com/datasets/fanconic/skin-cancer-malignant-vs-benign (accessed on 20 May 2022).

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
