# Peer review of "Classification of Skin Cancer Lesions Using Explainable Deep Learning"

_sensors, 2022, doi:10.3390/s22186915_

Round 1
Reviewer 1 Report
The application of deep learning in skin cancer diagnosis is admirable. The authors compared the accuracy of two models, MobileNetV2 and DenseNet201, and used GradCam to visualize the results. Dermatologists will be able to diagnose skin cancer with the method, however I have four reservations to address.
As two deep learning models were used in this study, their accuracy was high while they differed slightly. Ideally, there should be a great deal of overlap between the two models. This overlap part of the result should be very reliable, while the rest of the result may require further investigation by experts. They should analyze the positive intersection result from two models.
Grad-cam can only locate the dog, not the cat, as shown in figure 6. Does this method depend on the size of the target? If the localization will be affected by the tumor size, it is better to normalize the image based on the tumor size.
Since both models are in Colab, the authors may consider making them public. By doing this, dermatologists will have better access to those tools.
In line 272, the abbreviation of grad-cam was explained. It is better to put this sentence at the top, where the abbreviation appears first.
Author Response
Please kindly check the attached response file.

Reviewer 2 Report
Thank you for the opportunity of revising the manuscript entitled: „Classification of Skin Cancer Lesions Using Explainable Deep Learning”.
It is quite interesting from computer and cyber science view however the whole introduction section is full with elementary errors, e.g. malignant tumors DO NOT spread to other body organs through the epidermis layer of the skin, but through lymphatic vessels to lymphatic nodes at first, there is also a possibility to developed distant metastasis through blood vessels.
The are some mistakes in skin cancer classification, please check and cite the article: “Detailed head localization and incidence of skin cancers” Sci Rep 2021; 11(1): 12391 doi: 10.1038/s41598-021-91942-5.
The Authors write all the time that dermatologists deal with skin cancers. This is of course true but they are not only specialists who take care of patients with skin cancer but also surgeons, plastic surgeons, oncologist surgeons. I am aware that Authors are not medical personnel that is why I think it is mandatory that introduction section should be read by medical doctor and then sent for revision.
What are the medical possibilities of usage of this algorithm in doctors’ daily practice?
Is it going to be some software or application that can be buy?
In most cases when skin lesion is recognized eventually it is going to be excised and send to histopathological examination…
Please explain what benefits can be taken from your deep learning as it does not substitute pathological verification.
Author Response

(The authors gave the same response as above.)

Round 2
Reviewer 2 Report
The Authors introduced all remarks, in current form I recommend this article for publication.